# Theoretical Investigations for Strain Relaxation and Growth Mode of InAs Thin layers on GaAs(111)A †

**Tomonori Ito \*, Toru Akiyama and Kohji Nakamura**

Department of Physics Engineering, Mie University, Tsu 514-8507, Japan; akiyama@phen.mie-u.ac.jp (T.A.); kohji@phen.mie-u.ac.jp (K.N.)

**\*** Correspondence: tom@phen.mie-u.ac.jp; Tel.: +81-59-231-9724

† This paper is an extended version of our paper published in the 13th International Conference on Atomically Controlled Surfaces, Interfaces and Nanostructures (ACSIN2016), Rome, Italy, 9–15 October 2016.

**Abstract:** The growth mode of InAs/GaAs(111)A is systematically investigated using our macroscopic theory with the aid of empirical potential calculations that determine parameter values used in the macroscopic theory. Here, stacking-fault tetrahedron (SFT) found in InAs/GaAs(111)A and misfit dislocation (MD) formations are employed as strain relaxation mechanisms. The calculated results reveal that the MD formation occurs at the layer thickness $h$ about 7 monolayers (MLs). Moreover, we found that the SFT forming at $h$ about 4 MLs makes surface atoms move upward to reduce the strain energy to promote the two dimensional (2D) growth. Therefore, the SFT in addition to the MD plays an important role in strain relaxation in InAs thin layers on GaAs(111)A. The macroscopic free energy calculations for the growth mode imply that the InAs growth on the GaAs(111)A proceeds along the lower energy path from the 2D-coherent ($h \leq 4$ MLs) to the 2D-MD ($h \geq 7$ MLs) via the 2D-SFT (4 MLs $\leq h \leq 7$ MLs). Consequently, the 2D growth on the InAs/GaAs(111)A results from strain relaxation due to the formation of the SFT near the surface and the subsequent MD formation at the interface.

**Keywords:** computer simulation; growth mode; strain relaxation; stacking-fault tetrahedron; misfit dislocation; InAs/GaAs(111)A

## 1. Introduction

Low-dimensional nanostructures have received much attention from the scientific and engineering viewpoints because of their small size and large surface-to-volume ratios. In particular, the InAs/GaAs system is crucial for fabricating semiconductor nanostructures such as quantum dots (QDs) and stacking-fault tetrahedrons (SFTs). Due to the 7% lattice mismatch between InAs and GaAs, the InAs on the GaAs(001) produces three-dimensional (3D) QDs with Stranski–Krastanov (SK) growth mode [1]. Despite a constant lattice mismatch, the InAs on the GaAs(111)A with Ga topmost layer exhibits two-dimensional (2D) growth with the misfit dislocation (MD) formation [2,3]. Although many studies have been done investigating the QD formation on the InAs/GaAs(001) [4–12], there have been only a few studies of the relationship between strain relaxation and the 2D growth on the InAs/GaAs(111)A [3,13,14]. Strain relaxation in the InAs/GaAs(111)A heteroepitaxy has been observed by scanning tunnelling microscopy (STM), where the MD network formation is identified as a strain relaxation mechanism occurring at layer thickness $h$ = 3 monolayers (MLs) [3]. On the basis of the model observed by STM, energy calculations using the valence force field (VFF) model clarify that the semicoherent interface consisting of a network of intersecting MDs at $h \geq 4$ MLs fully relieves the strain at the interface [13]. Rocking-curve analysis of reflection high-energy electron diffraction (RHEED) reveals that elastic distortion of InAs lattice during layer-by-layer growth is found only

below 1.5 MLs. Moreover, the RHEED analysis also indicates that strain in the direction parallel to the surface drastically relaxed in the range of 1.5 MLs $\leq h \leq$ 3 MLs, while its lattice constant gradually approaches that of bulk InAs at $h \geq$ 3 MLs [14]. Although these results are consistent in some aspects, the strain relaxation process depending on layer thickness is still unclear. In this study, the strain relaxation process and resultant growth mode in the InAs/GaAs(111)A are systematically investigated using our macroscopic theory with the aid of empirical potential calculations to determine parameter values used in the macroscopic theory. Here, the formation of the SFT is employed as a strain relaxation mechanism in addition to the MD formation. The SFT with nano- to micro-meter size is often found in InAs layers on GaAs(111), where the SFT is surrounded by the (111)A surface and three triangular {111}-stacking-fault planes consisting of wurtzite structure below the surface [15].

## 2. Computational Methods

Schematics of the computational models considered in this study are shown in Figure 1, including (a) 2D growth with MD (2D-MD) and (b) 2D growth with SFT (2D-SFT), except the model for 2D coherent growth (2D-coherent). The MD core with five- and seven-member rings (5/7 core) are inserted at the interface between InAs and GaAs(111), as shown in Figure 1a. The 5/7 core is often found in transmission electron microscopy (TEM) observations [16], and is recognized to be the stable core structure in the MD formation energy calculations for compound semiconductors [4,17,18]. In Figure 1b, the SFT consists of the face with stacking-fault and the ridge corresponding to stair–rod dislocation along the (110) direction, similarly to the SFT in Si [19]. According to our macroscopic theory for growth mode [20], the free energies $F$ for these growth modes are given by

$$F_{\text{2D-coh}}(h) = \gamma + M\varepsilon_0{}^2 h/2, \tag{1}$$

$$F_{\text{2D-MD}}(h) = \gamma + E_{\text{d}}/l_0 - E_{\text{d}}{}^2/2(M\varepsilon_0{}^2 l_0{}^2 h), \tag{2}$$

$$F_{\text{2D-SFT}}(h) = \gamma + M\varepsilon_0{}^2 h/2 + E_{\text{SFT}}/A_{\text{unit}}, \tag{3}$$

where $\gamma$ is the surface energy, $M$ the effective elastic constant, $\varepsilon_0$ the intrinsic strain (= 0.072), $h$ the layer thickness, $E_{\text{d}}$ the MD dislocation formation energy, $l_0$ the average dislocation spacing (= 58.76 Å), $E_{\text{SFT}}$ the SFT formation energy, and $A_{\text{unit}}$ the area of 14 × 14 planar unit cell.

In order to obtain the parameter values of $M$, $E_{\text{d}}$, and $E_{\text{SFT}}$, we employ a simple formula for estimating system energy $E$ for computational models such as the 2D-coherent, the 2D-MD, and the 2D-SFT as follows.

$$E = E_0 + \Delta E_{\text{SF}}, \tag{4}$$

$$E_0 = 1/2 \times \sum_{i,j} V_{\text{ij}}, \tag{5}$$

$$V_{\text{ij}} = A\exp[-\beta(r_{\text{ij}} - R_{\text{i}})^\gamma][\exp(-\theta r_{\text{ij}}) - B_0\exp(-\lambda r_{\text{ij}})G(\eta)/Z_{\text{i}}{}^\alpha], \tag{6}$$

$$\Delta E_{\text{SF}} = K[3/2 \times (1 - f_{\text{i}}) \times Z_{\text{b}}{}^2/r_{\text{bb}} - f_{\text{i}} \times Z_{\text{i}}{}^2/r_{\text{ii}}]. \tag{7}$$

Here, $E_0$ is the cohesive energy estimated by Kohr–Das Sarma-type empirical interatomic potential $V_{\text{ij}}$ within the second-neighbor interactions [21]. The potential parameters $A$, $\beta$, $R_{\text{i}}$, $\gamma$, $\theta$, $B_0$, $\lambda$, $\eta$, and $\alpha$ are determined by reproducing equilibrium interatomic bond lengths, elastic stiffness for zinc blende structure, and relative stability among zinc blende, rocksalt, and CsCl structures [22]. Stacking-fault energy $\Delta E_{\text{SF}}$ is described as the summation of electrostatic energies consisting of repulsive interaction between covalent bond charges $Z_b$ (= −2) and attractive interaction between ionic charges $Z_i$ (= ±3 for III-V compound semiconductors) depending on ionicity $f_{\text{i}}$ (= 0.298 for InAs) [23]. The value of coefficient $K$ is determined to be 8.7 (meV·Å) by reproducing energy difference 25.3 (meV/atom) between diamond and hexagonal structures for C with $f_{\text{i}}$ = 0 obtained by ab initio calculations. In the system energy calculations, the lattice parameter $a$ of InAs is fixed to be that of GaAs(111) for the 2D-coherent and the 2D-SFT or relaxed value of $a$ for the 2D-MD, and the lattice parameter $c$ and the

atomic positions are varied to minimize the system energy in a $14 \times 14$ planar unit cell with increase of layer thickness $h$.

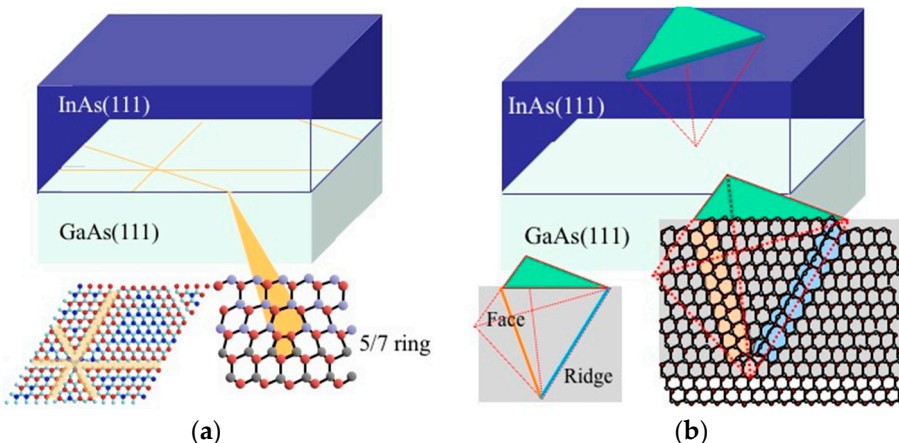

**Figure 1.** Schematics of the computational models for the InAs/GaAs(111)A system considered in this study: (**a**) 2D growth with dislocation (5/7-ring core structure) network formation at the interface; (**b**) 2D growth with formation of stacking-fault tetrahedron (SFT) consisting of ridge and face.

## 3. Results and Discussion

Figure 2a shows the calculated energy difference $\Delta E_{MD}$ between the 2D-coherent and the 2D-MD as a function of layer thickness $h$. This reveals that the $\Delta E_{MD}$ changes its sign from positive to negative at $7 \leq h \leq 8$ MLs, where the strain in InAs thin layers is relaxed to stabilize the 2D-MD. The critical layer thickness $7 \leq h \leq 8$ MLs for the MD generation agrees well with its value of $h$ about 7 MLs estimated from People-Bean's formula to minimize energy in the entire crystal at thermodynamic equilibrium [24]. Using the calculated results shown in Figure 2a, the parameter values used in the free energy formula are determined to be $M = 2.63 \times 10^{10}$ (N/m$^2$) and $E_d = 0.675$ (eV/Å). The calculated energy difference $\Delta E_{SFT}$ between the 2D-coherent and the 2D-SFT as a function of layer thickness is shown in Figure 2b, where the $\Delta E_{SFT}$ becomes negative at $h \geq 4$ MLs. This suggests that the SFT formation acts as a strain relaxation mechanism near the surface as well as the MD formation at the interface in InAs/GaAs(111)A system. The $\Delta E_{SFT}$ results from the competition between energy profit in the face region and energy deficits in the face and the ridge regions. The face region dramatically decreases the system energy due to strain relaxation, inducing upward displacements of atoms in the SFT that overwhelm the energy deficit due to the stacking-fault formation. In the ridge region, however, the stair–rod dislocation increases system energy due to its energetically unfavorable dimers along the ridge line, shown in Figure 1b. The calculated results shown in Figure 2b approximately determine the parameter value of $E_{SFT} = 0.014h - 0.0011h^2$ (eV) as a function of layer thickness $h$.

Figure 3a depicts the calculated free energy differences $\Delta F$ between the 2D-coherent and various growth modes such as the 2D-MD and the 2D-SFT as a function of layer thickness $h$. It should be noted that the $\Delta F$ for the 2D-MD becomes negative at $h$ about 6 MLs different from $7 \leq h \leq 8$ MLs shown in Figure 2a. This means that the first isolated dislocation formation occurs at $h \sim 6$ MLs, and the dislocation spacing $l$ gradually decreases according to $l = l_0/(1 - h_c^{MD}/h)$ approaching the average dislocation spacing $l_0$ (= 58.76 Å) with increase of $h$, where $h_c^{MD}$ is estimated by $E_d/(M\varepsilon_0^2 l_0)$ [20]. Figure 3a implies that the InAs growth on the GaAs(111)A proceeds along the lower energy path from the 2D-coherent ($h \leq 4$ MLs) to the 2D-MD ($h \geq 7$ MLs) via the 2D-SFT (4 MLs $\leq h \leq 7$ MLs). This is consistent with STM observations, where faulted triangle domains containing stacking-faults appear with the MD network at 5 MLs on the InAs/GaAs(111) [3]. Furthermore, a similar process was found in molecular dynamics simulations for (111)-oriented heteroepitaxial Al films forming a local disorder zone like the SFT near surface layers followed by the MD nucleation [25]. Experimental results,

however, indicate that strain is gradually relieved beyond 1–2 MLs, contradicting our calculated results with 4 MLs [3,14]. This discrepancy can be interpreted by considering the fact that our computational model is not optimized for 2D-SFT. The 14 × 14 unit cell used in this study is not set for the SFT, but for the MD with geometrically optimized dislocation spacing. If the SFT spacing is optimized, the $\Delta E_{SFT}$ becomes lower to give smaller layer thickness for the SFT formation. Moreover, employing the MD consisting of faulted and unfaulted domains observed by STM [3], further strain relaxation may occur at smaller layer thickness [13]. Consequently, our calculated results suggest that the first strain relaxation occurs near the surface due to the SFT formation before the MD formation at the interface.

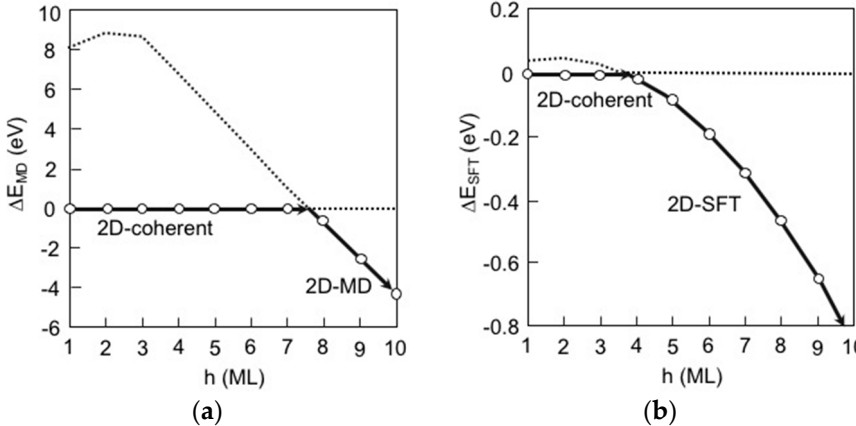

**Figure 2.** Calculated energy differences as a function of layer thickness $h$: (**a**) $\Delta E_{MD}$ between the 2D coherent growth (2D-coherent) and the 2D growth with misfit dislocation (2D-MD); (**b**) $\Delta E_{SFT}$ between the 2D-coherent and the 2D growth with SFT (2D-SFT).

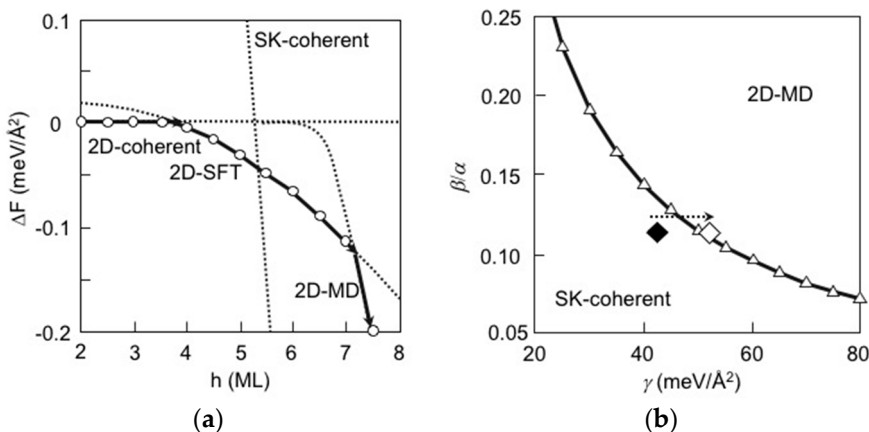

**Figure 3.** (**a**) Calculated free energy differences $\Delta F$ as a function of layer thickness $h$, where the $\Delta F$ for the Stranski–Krastanov (SK)-coherent is also shown in this figure; (**b**) Calculated growth mode boundary between the 2D-MD and SK-coherent as functions of $\gamma$ and $\beta/\alpha$. The closed diamond denotes the data extracted from [6].

In order to discuss the growth mode on the InAs/GaAs(111) in more detail, we incorporate a three-dimensional (3D) coherent growth mode (the SK-coherent). To this end, we employ the free energy of the SK-coherent as follows [20]:

$$F_{SK\text{-}coh}(h) = \gamma(1 + \beta) + M(1 - \alpha)\varepsilon_0{}^2 h/2, \tag{8}$$

where $\beta$ and $-\alpha$ are the effective energy increase in surface energy of the epitaxial layer and the effective decrease in strain energy due to SK-island formation. Assuming $\gamma$ = 42 (meV/Å$^2$) for the InAs(111)A, $\beta$ = 0.084, and $\alpha$ = 0.748 extracted from the previously reported results for the InAs/GaAs(001) [6], the calculated free energy difference $\Delta F$ between the 2D-coherent and the SK-coherent is also shown in Figure 3a. Although the $\Delta F$ for the SK-coherent becomes negative at 5 MLs $\leq h \leq$ 6 MLs that is energetically competitive with the 2D-MD, the SK-coherent does not appear due to the 2D-SFT preceding the SK-coherent. Using Equations (2) and (8), the growth mode boundary between the 2D-coherent and the SK-coherent is simply described by the following equation:

$$\beta/\alpha = (E_d/l_0)/(2\gamma) = 5.744/\gamma. \tag{9}$$

The calculated growth mode boundary is shown in Figure 3b according to Equation (9). The closed diamond denotes the data with $\gamma$ for the InAs(111)A, and $\beta/\alpha$ extracted from the results for the InAs/GaAs(001).

It is found that the SK-coherent mode is more favourable than the 2D-MD, consistent with Figure 3a. However, there have been some reports where the strain increases InAs surface energy by 10–20 (meV/Å$^2$) [6]. It should be noted that the increase of surface energy tends to prefer the 2D-MD exemplified by open diamond with $\gamma$ = 52 (meV/Å$^2$) keeping $\beta/\alpha$ constant, as shown in Figure 3b. Although further study of surface energy depending on strain is necessary to discuss the growth mode more precisely (including the SK-coherent), the 2D growth is preferable on the InAs/GaAs(111)A, resulting from the strain relaxation forming the SFT near the surface followed by the formation of MD at the interface.

## 4. Conclusions

We have theoretically investigated the strain relaxation and the resultant growth mode on the InAs/GaAs(111)A using our macroscopic theory. SFT formation plays an important role in the strain relaxation near the surface, leading to 2D growth on the InAs/GaAs(111)A. In conclusion, the InAs growth proceeds from the 2D-coherent to the conventional 2D-MD via the 2D-SFT without the SK-coherent.

**Acknowledgments:** This work was partially supported by JSPS KAKENHI Grant Number 16K04962.

**Author Contributions:** Tomonori Ito conceived and designed the calculationsToru Akiyama performed the calculations; Kohji Nakamura analyzed the data. All authors have read and approved the final manuscript.

**Conflicts of Interest:** The authors declare no conflict of interest.

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
