# Peer review of "Theoretical Investigations for Strain Relaxation and Growth Mode of InAs Thin layers on GaAs(111)A"

_condensedmatter, doi:10.3390/condmat1010004_

Round 1

Reviewer 1 Report

Minor comments:

1) Authors should define meaning of A in GaAs(111)A early in the manuscript

2) Figure 1 could be improved. It was unclear how the atomic structure in Fig 1b related to the SFT. The authors seem to be trying to use yellow and gray colors, but this was confusing.

Author Response

Dear Editor

We thank the referee for describing our paper as suitable for this journal. We have made the following changes:

1) We define the meaning of "A" as " GaAs(111)A with Ga topmost layer".

2) We include detail atomic arrangements for the SFT as a cross sectional view in Fig. 1b. 

3) We change the title by deleting "resultant". This is because ACSIN2016 Conference Chair Prof. Bianconi asked us that the title of the paper for MDPI Condensed Matter should be different the published extended abstract published in the Book of Abstract after having submitted our paper. 

4) Moreover, Prof. Bianconi asked us to add the text such that the paper has been presented at ACSIN2016 and the reference for the Book of Abstract. We add the text at Line 172-175 and reference [26].

Sincerely,

Tomonori Ito

Reviewer 2 Report

In this article, the authors investigated the growth mode of the InAs/GaAs(111)A with their macroscopic theory. They found that both stacking-fault tetrahedron (SFT) and misfit dislocation (MD) formations play important roles during InAs thin layers growth on GaAS(111)A.The InAs growth proceeds along the lower energy path from the 2D-coherent, the 2D-SFT, to the 2D-MD without 3D coherent mode, which is consistent with Scanning tunneling microscopy (STM) observations. This work is very useful for studying the growth of materials. A few points need attention:

Some mistakes of writing: for example, Line 37, tunnelling; Line 17, 51, grown. Some sentences such as 10-12, 20-22, 50-52 are not easily understood.

Citations are needed after InAs/GaAs(111)A (Line 36).

Author Response

Dear Editor

We thank the referee for describing our paper as a good paper and for his suggestions, we have made the following changes.

1) We corrected mistakes of writing following the referee's suggestions.

2) We add references [3,13,14] after InAs/GaAs(111)A.

3) We change the title by deleting "resultant". This is because ACSIN2016 Conference Chair Prof. Bianconi asked us that the title of the paper for MDPI Condensed Matter should be different the published extended abstract published in the Book of Abstract after having submitted our paper. 

4) Moreover, Prof. Bianconi asked us to add the text such that the paper has been presented at ACSIN2016 and the reference for the Book of Abstract. We add the text at Line 172-175 and reference [26].

Sincerely,

Tomonori Ito